# Mesoporous Bioglasses Enriched with Bioactive Agents for Bone Repair, with a Special Highlight of María Vallet-Regí’s Contribution

**DOI:** 10.3390/pharmaceutics14010202

**Published:** 2022-01-15

**Authors:** Antonio J. Salinas, Pedro Esbrit

**Affiliations:** 1Departamento Química en Ciencias Farmacéuticas, Facultad de Farmacia, Universidad Complutense de Madrid, UCM, Instituto de Investigación Hospital 12 de Octubre, imas12, 28040 Madrid, Spain; pesbrit@gmail.com; 2Networking Research Center on Bioengineering, Biomaterials and Nanomedicine, CIBER-BBN, 28040 Madrid, Spain

**Keywords:** mesoporous bioactive glasses, Prof. Vallet-Regí: regenerative medicine, bone repair, therapeutical ions, bioactive biomolecules, stem cells

## Abstract

Throughout her impressive scientific career, Prof. María Vallet-Regí opened various research lines aimed at designing new bioceramics, including mesoporous bioactive glasses for bone tissue engineering applications. These bioactive glasses can be considered a spin-off of silica mesoporous materials because they are designed with a similar technical approach. Mesoporous glasses in addition to SiO_2_ contain significant amounts of other oxides, particularly CaO and P_2_O_5_ and therefore, they exhibit quite different properties and clinical applications than mesoporous silica compounds. Both materials exhibit ordered mesoporous structures with a very narrow pore size distribution that are achieved by using surfactants during their synthesis. The characteristics of mesoporous glasses made them suitable to be enriched with various osteogenic agents, namely inorganic ions and biopeptides as well as mesenchymal cells. In the present review, we summarize the evolution of mesoporous bioactive glasses research for bone repair, with a special highlight on the impact of Prof. María Vallet-Regí´s contribution to the field.

## 1. Rationale and Objectives

Writing a review article in tribute to María Vallet-Regís’s scientific career is a colossal task. From the beginning of her excellent scientific career, her research interest has been consistently on the frontier of knowledge, opening new research lines followed by many other colleagues. Her training as a solid-state chemist led her to evolve from an initial research interest in non-structural materials such as pigments, magnets and high-temperature superconductors, to focus on materials for biomedical applications in the last few years. In the latter area, she has investigated different types of biomaterials from inert or bioactive ceramics, including calcium phosphates, glasses and glass ceramics, to organic–inorganic hybrid materials and silica-based mesoporous materials. In the latter type of materials, she pioneered proposing their use as a matrix for drug delivery systems [1,2] and obtaining nanoparticles for applications in nanomedicine [3].

In the context of the Special Issue of Pharmaceutics in tribute to María Vallet-Regís’s scientific career since she proposed the application of silica-based materials in the biomedical field, the present review is devoted to mesoporous bioactive glasses (MBGs). The latter is a category of bioceramics that, as will be explained, can be considered intermediate between traditional bioactive glasses—obtained by quenching of a melt or by the sol-gel method—and silica mesoporous materials. We will highlight the specific research achievements of María Vallet-Regí and her inspirational task for other expert colleagues in the MBGs field. This review focuses on all the articles published by María Vallet-Regí et al. on bioactive mesoporous glasses, namely those containing CaO or P_2_O_5_ together with SiO_2_ in their composition, obtained by using surfactants as structure-directing agents. Therefore, we include about 55 research articles—representing about 7% of her scientific production—coauthored by María Vallet-Regí where MBGs played a pivotal role. To add context to these contributions, outstanding references by other authors on the same topic were also included.

## 2. The Need of Synthetic Biomaterials in Bone Regeneration

Bone regeneration remains an important challenge in orthopedic surgery [4,5,6]. In this regard, bone tissue engineering aims to promote the self-regeneration of bone tissue injury [7]. This can be performed by using a wide range of biocompatible materials combined or not with osteogenic cells, which provide physical support and suitable biochemical signals for promoting bone healing [8,9,10]. These bioactive materials are paramount in this regard to repair osteoporotic bone exhibiting low bone mineral density and bone fragility, which hampers the use of more rigid metal-based implants. These materials are commonly fabricated as a three-dimensional structure, termed “scaffold” exhibiting high porosity and pore interconnectivity to host various osteogenic agents to promote osteoblastic growth and function [11,12].

During the last decades, a variety of biomaterials as synthetic bone graft substitutes have been synthesized. In this respect, glass-ceramics such as silica-based glasses containing Si, Ca and P are particularly interesting for their ability to form a calcium phosphate layer that enables binding to bone and prevents the formation of a fibrous layer around the material used as an implant. An early fixation and well osseointegration of the implant is highly demanded during surgical practices since implant instability increases the risk of aseptic loosening. Ca and P are the main components of bone apatite (Ca_10_(PO_4_,CO_3_)_6_(OH)_2_), and thus they have a key role in bone remodeling [13,14]. Both Si and Ca ions can improve osteogenesis by upregulating the expression of osteogenic genes, and are associated with bone tissue mineralization [15,16,17,18]. These bioactive ceramics also induce vascularization [19,20], and can interact and act in concert with osteogenic cells to promote bone formation, and as such are recognized for their excellent biocompatibility and osteoconductivity, as well as for their antimicrobial properties [21,22]. They can be manufactured (e.g., as the inorganic component in organic–inorganic hybrid materials) with improved mechanical properties to provide suitable physical support during bone healing [7,23,24].

In this scenario, MBGs with a highly ordered network of pores between two and ten in diameter [25,26] are of particular interest and will deserve special consideration in the present review. The interest in developing MBG as implant scaffolds comes from the need to provide a suitable microenvironment to favor bone regeneration, which is achieved by gradual removal of the scaffold (biodegradation) and its replacement by new bone tissue at the implant site [27].

## 3. The Three Families of Bioactive Glasses

In the 1970s, it was discovered that some glass compositions as implants failed to elicit a foreign body reaction [28]. Accordingly, these glasses, currently denoted as bioactive glasses (BGs), were not found to be surrounded by a fibrous capsule, but integrated into the host by forming a mechanically strong bond with the adjacent living tissue. This finding opened new avenues in the design of biomaterials mainly in Orthopedics and Dentistry [29]. Since the early stages of BGs development, it was evidenced that those with the faster bioactive response have a capacity to bind to a variety of soft tissues; justifying their current increasing interest for clinical applications beyond the skeletal system [30].

However, melt-prepared BGs (MPGs), found only rather few clinical applications, namely in the dental field, for replacing the middle ear chain ossicles, or as particulates in bone grafts, in the last twenty years [29]. For this reason, in the 1990s, a new family of BGs, the so-called bioactive sol-gel glasses (SGGs), was obtained by wet chemistry methods, exhibiting high porosity and specific surface area [31]. These interesting textural properties led to the assumption that SGGs could be used as matrices in drug delivery systems. Furthermore, the sol-gel method allowed processing BGs as coatings or fibers, which are shapes difficult to obtain with the traditional quenching of a melt method. However, SGGs did not meet all the initial expectations for clinical applications because their bioactive response was analogous to MPGs and because due to their large pore size dispersion they did not allow efficient control of the release of biologically active substances. Indeed, in SGGs the bioactivity window was expanded because they were coated by an apatite-like layer after soaking in a Simulated Body Fluid (SBF), even for silica contents of 90%; while such response was limited to a maximum content of 60% for MPGs. These findings led to the assumption that SGGs might present a quicker bioactive response in SBF than MPGs. However, this was not the case; a similar period of time (2 to 7 days) was necessary to form the apatite-like layer in both types of glasses [25]. In the 1990s, while SGGs were developed, in the field of catalysis, mesoporous silica materials were designed to obtain materials with homogeneous pore sizes larger than that of zeolites. They were obtained in the presence of a surfactant, producing an ordered mesoporous arrangement with a very narrow pore size distribution [32]. The very high specific surface areas and pore volumes together with high uniformity in pore size make these materials ideal materials for the controlled release of biomolecules and drugs, as it was proposed by Vallet-Regí et al. for pure silica mesoporous materials [1]. This spurred the development of newly designed BGs, early named glasses with template glasses (TGs) and later mesoporous bioactive glasses (MBGs). 

MBGs exhibit intermediate properties between SGGs and pure silica mesoporous materials. Indeed, the synthesis process of the MBGs is based on sol-gel processing similar to that of SGGs, but including a surfactant acting as a structure-directing agent as is used to obtain pure silica mesoporous materials such as MCM-41 or SBA-15. Unlike mesoporous SiO_2_-based materials, however, MBGs also contain other oxides such as CaO, or P_2_O_5_. The initial attempts to obtain MBGs composed of silicon, calcium and phosphorus oxides by the synthesis method of silica mesoporous materials failed because the procedure is performed at a pH that provokes calcium precipitation as calcium hydroxide. However, such synthesis can be carried out by using a method proposed by Brinker et al. [33], the so-called evaporation induced self-assembly (EISA) method, where the solvent evaporation produces the required increase in the surfactant concentration until reaching the critical micellar concentration where the ordered mesophase is formed. In this way, the research groups led by Zhao [34] and Vallet-Regí [35], independently, opened a new avenue for the synthesis and characterization of this exciting family of BGs.

Figure 1A shows the main features of glasses which are amorphous materials with disordered structures at the atomic scale. As observed, glasses are formed by two types of oxides: elements with high oxidation state, such as Si (IV), P(V) or B(III), as the network formers; and elements with low oxidation state, such as alkaline (Na^+^, K^+^) and alkaline-earth (Ca^2+^, Mg^2+^) as network modifiers. These formers give glass stability under atmospheric conditions, confer specific properties and decrease the temperature of the synthesis process. Figure 1B, shows the most distinctive features of three families of BGs including the year of their initial synthesis.

The textural properties, i.e., surface area and pore volume, of the three families of glasses, MPG, SGG and MBG, are compared in Figure 2. As observed, MPGs are dense, non-porous materials with a minimum value of the surface area. On the other hand, SGGs and MBGs are both porous materials, but in the latter material, the mesopores are all of the same diameters and appear ordered. Moreover, the textural properties (surface area and pore volume) of MBGs are roughly twice those of SGGs. This feature is considered to be responsible for the greater bioactive kinetics of MBGs compared to SGGs and MPGs [36]. TEM images of the SGG and of the MBG shown in the figure correspond both to glasses of the SiO_2_–CaO–P_2_O_5_ system. The difference obtained lies in the synthesis method, the former was performed by the classical sol-gel method and the latter in the presence of a surfactant using the solvent evaporation induced self-assembly (EISA) method. As shown in the figure, mesopores of the SGG are disordered and exhibit different sizes. In contrast, an ordered arrangement of pores of the same size is observed in the MBG micrograph.

Figure 3 shows how the timeline development of the new families of BG has improved their properties for use in bone regeneration. MPG are excellent biomaterials for bone graft substitution due to their quick bioactive response, the capacity of hosting therapeutic inorganic ions and the possibility of their processing to obtain scaffolds and composites. In addition to the latter properties, SGGs offer other interesting features arising from their wet chemistry processing methods at low temperatures. First, the bioactivity improvement by including SiO_2_ (up to 90 mol-%) as a result of their excellent textural properties. Wet chemistry methods were introduced in search of a surface richer in silanol groups but also resulted in glasses with high specific surface area and porosity. Furthermore, SGGs can be functionalized and biocompatible polymers can be added during their synthesis to obtain organic–inorganic hybrid materials (nanocomposites) [37,38,39] with the desired mechanical or degradation properties. Additionally, by selecting the appropriate time during the sol to gel transition, it is possible to obtain coatings or fiber meshes of SGGs.

MBGs, which can be considered an improvement of SGGs, have all the characteristics of MPGs and SGGs, but present other new features as a result of their synthesis in the presence of surfactants, producing a greater control of their mesostructure. This process produces advantageous textural properties as well as a much faster in vitro bioactive response than that of the other BGs family members [11]. The large volume of monodisperse pores makes MBGs ideal candidates for hosting different molecules, which is essential for their putative applications in bone tissue engineering. However, the previously mentioned EISA method for MBG synthesis yields powders with limited biomedical applications, mainly as bone fillers, and other biomaterials of natural or synthetic origin are currently available as bone grafts in this regard. In contrast, processing MBG as 3D porous scaffolds with hierarchical porosity of different size scales has more interest for possible applications in regenerative medicine. Moreover, one of the currently most active areas of MBGs research concerns their use as nanocarriers of biologically active ions and therapeutic biomolecules [40,41]. This requires their synthesis as nanoparticles of variable size between 20 and 800 nm, and with a pore size tailored in the range 5–20 nm. However, this kind of MBG form still needs a great deal of research before it could be materialized into a clinical application. Indeed, there is still scarce preclinical research in animal models with MBGs. The current results are very promising, but several aspects still need to be elucidated before considering their putative clinical use: the ideal composition of the glass, the optimal resorption rate, the type and amount of therapeutic ions included, or the addition of different bioactive drugs presenting osteogenic, bactericidal, angiogenic, anticancer, anti-inflammatory, antioxidant and/or antiviral properties.

## 4. Bioactive Glasses in Bone Regeneration

In the 1980s, BGs based on the SiO_2_–CaO–P_2_O_5_–Na_2_O system were approved by the FDA for several maxillofacial and dental applications [42]. Later on, their applications were extended to Orthopaedics as bone grafts in non-load bearing sites. Recent developments in the performance of these materials rely on new methods of synthesis to obtain materials specifically depicting an ordered mesoporosity, as described above [34,35]. Of interest in this respect, the structural and bioactive properties of MBGs provide their ability for loading a variety of osteogenic agents to improve their use as scaffolds in bone regeneration [43,44,45]. Increasing evidence indicates that ionic dissolution from these glasses is key in their osteogenic behavior [46]. Indeed, a variety of trace elements (Sr, Cu, Zn) present in the human body have anabolic bone activity [47,48]. Accordingly, MBGs based on SiO_2_–CaO–P_2_O_5_ and containing distinct cations were designed according to the required clinical applications [29,49,50]; this includes a recently reported P_2_O_5_-free borate glass porous scaffold with a high Ca content which shows both osteogenic and antimicrobial features [12].

Sr is one of the various cations incorporated into ceramic biomaterials due to its well-characterized osteogenic properties: it has a dual role in inducing osteoblastogenesis by stimulating the osteogenic differentiation program while inhibiting adipogenesis from the common mesenchymal progenitor cell and decreasing osteoblast apoptosis in vitro. This is of particular interest when considering the high adiposity often observed in osteoporotic bone. In fact, strontium ranelate—an approved osteoporosis therapy—was shown to improve trabecular bone structure and prevent fracture in postmenopausal women [51,52,53]. This cation appears to act, at least in part, through the CaSR in osteoblasts [43]. Nanohydroxyapatites co-substituted with foreign ions such as Mg^2+^ and CO_3_^2−^ to mimic the composition of bone HA, and also Sr^2+^ as an osteogenic signal and porous Sr-substituted calcium silicate ceramic scaffolds were developed as biomaterials to regenerate osteoporotic bone [54,55]. Moreover, using a canine gap model, incorporation of 5% Sr to an HA bone graft favored its fixation followed by healing of the gap possibly related to the anabolic and anticatabolic effects of the cation [56]. In addition, adding Sr to either MBG scaffolds fabricated using a 3D printing method or bioactive glass/polycaprolactone (a resorbable polymer widely used in bone tissue engineering) composite scaffolds enhanced the osteogenic activity of the respective biomaterial without the cation [57,58]. Additionally of note, the incorporation of Sr to borate glass improved its bone growing action [59]. Sr-stabilized bulk glass ceramics as the implant was shown to improve the healing of a bone defect in a rabbit model [60]. Thus, Sr is being extensively used to dope various biomaterials including bioactive BGs as bone tissue engineering strategies for bone healing.

Cu is an essential element whose deficit produces osteopenia in humans [61,62]. This was related in part to the well-known role of Cu on the activity of antioxidant enzymes involved in the cross-linking of bone collagen fibers [63]. Moreover, Cu has antimicrobial activity [64,65]. Recently, MBGs containing 2.5–5% CuO, synthesized using HNO_3_ as catalyst and calcium and copper nitrates as CaO and CuO precursors, respectively, showed a rapid bioactive response (formation of an apatite-like layer) in simulated body fluid in vitro; an activity often related to the in vivo mineralization process [66,67]. Moreover, the capacity of these modified MBGs to release supraphysiological amounts of Ca^2+^ and Cu^2+^ ions and to host osteogenic agents in their pores make them interesting biomaterials for bone regeneration.

Zn is known to have angiogenic, osteogenic and bactericidal properties [68,69,70]. Recently, MBGs of composition 80–x%SiO_2_–15%CaO–5%P_2_O_5_ with 4–5% ZnO and manufactured as disks, were evaluated in pre-osteoblastic MC3T3-E1 cell cultures. Some of these materials were also loaded with the osteoinductive peptide osteostatin, a C-terminal peptide from parathyroid hormone-related protein [PTHrP (107–111)] [7]. Interestingly, the simultaneous presence of Zn^2+^ and osteostatin in these MBGs was found to potentiate their capacity to increase MC3T3-E1 cell growth and osteoblastic differentiation [71]. Additional studies were subsequently carried out to further explore the suitability of this approach in bone regeneration. Thus, MBGs of the same composition were fabricated as 3D scaffolds, which showed optimal hierarchical porosity (from mesopores to macrospores), specific surface area and in vitro bioactivity for putative use as bone substitutes [72]. Impregnation of these scaffolds with osteostatin enhanced their osteogenic capacity by promoting human mesenchymal cell (hMSC) colonization and proliferation. Furthermore, as observed with these materials as disks in MC3T3-E1 cell cultures, Zn^2+^ and osteostatin together in the scaffolds induced several osteoblast differentiation genes and mineralization in hMSCs without the addition of other osteogenic differentiation-promoting factors [72]. Therefore, osteostatin was confirmed to enhance the osteogenic capacity of Zn^2+^-enriched MBGs. 

As a preclinical approach, similar scaffolds [with composition (mol%) 82.2SiO_2_–10.3CaO–3.3P_2_O_5_–4.2ZnO were coated with gelatin that facilitates both their handling and the release of inorganic ions and peptides, and were then evaluated as implants in two rabbit models of bone regeneration. These scaffolds, containing or not osteostatin and hMSCs, were implanted into bone defects (7.5 mm diameter, 12 mm depth) drilled in the distal femoral epiphysis of New Zealand rabbits [73]. Three months thereafter, the presence of osteostatin and hMSCs in the implanted scaffold significantly improved bone healing by inducing implant degradation, reducing the fibrous cup observed with the raw scaffolds and increasing trabecular bone volume density [73]. These recent findings give credence to the notion that these functionalized MBG scaffolds might be considered as a new interesting approach in bone tissue engineering.

## 5. Timeline Evolution of MBG Research by Vallet-Regí´s Group

Since the first MBG report in 2004, the studies carried out by María Vallet-Regí´s group in this field for more than a decade may well represent the timeline of the evolution in the research interest of the BG community of investigators (Table 1). Thus, in this section, we will deal with these chronological studies in detail.

The earliest study by Vallet-Regí´s group [35] described the influence of the amount of surfactant Pluronic^®^ 123 on the material mesostructure and the very quick in vitro bioactive response of MBG. It was shown the formation of different ordered mesopores arrangements, from the 3D-cubic to the 2D hexagonal, highlighting their unusually high textual properties compared to those of the well-known SGGs. In the first review articles on this subject, this family of BGs was referred to as template glasses, because of the use of a surfactant acting as a template, that is a structure-directing agent. Nowadays, however, there is a wide consensus to denote these materials as MBGs.

The left side of Figure 4 shows the members of María Vallet-Regí*’*s group who have promoted the development of the MBGs, including the year of their first publication on the subject since that year can be considered as the kick-off of their research on these materials. We also included the names of the Ph.D. students whose Theses have focused exclusively on MBGs since, despite being trainees, their participation is considered very important in the described developments. They are: A. Lopez-Noriega, S. Shruti, M. Cicuéndez, N. Gómez-Cerezo, C. Heras, L. Casarrubios, J. Jiménez-Holguín. The right side of the figure shows the international and national research groups with which María Vallet-Regí has collaborated in MBGs research. In all cases, the results were published in high-impact journals indicating in the figure the year of the first joint publication. In some cases, the collaborations have produced several publications and some collaborations are still ongoing.

The same group [75] proceeded by publishing an in-depth study on these materials at an atomic scale based on solid-state NMR spectroscopy measurements to investigate various structural aspects, namely the location of phosphorus atoms in the structure and the impact of calcium excess. These studies provided a rationale to explain the higher bioactivity in the simulated biological medium of MBGs than MPGs or SGGs related to the very high textual properties and the thicker layer of silica gel formed on its surface in the former material [74]. In addition, the surface pH reaches a value of 7.4 in the case of MPGs and SGGs but is 6.7 in the case of MBGs. This lower pH in the latter accelerates the formation of octacalcium phosphate (OCP), which is the intermediate phase in the process of bone mineral maturation from amorphous calcium phosphate (ACP) to nano carbonate HA (n-CHA). Formation of OCP is never detected with the other aforementioned BGs.

Soon thereafter, Vallet Regí´s group [78,79] pioneered the synthesis and characterization of MBG microspheres to facilitate their possible use in bone grafting and drug release. During this initial period, their original studies on MBGs also included: incorporating phosphorus to mesostructured silica as a novel approach to reduce the leaching of silica; the essential role of calcium phosphate heterogeneities in the 2D hexagonal and 3D cubic structures; and synthesis and characterization of mesoporous microspheres with a doubly ordered core-shell structure [76,77]. Later on, this group has used alternative methods based on molecular models in silico—initially used to analyze the structure of simple ordered mesoporous materials containing silica, MCM-41 and SBA-15 —to unravel the structure and bioactivity of MBGs [123].

These studies were contemporary with studies in which MBGs were loaded with ipriflavone, an inhibitor of bone resorption, after functionalization post-synthesis with different organic groups to retain the drug into the mesoporous network, as a potential system for bone regeneration [80]. In addition, the group reported the use of solid-state nuclear magnetic resonance (NMR) to investigate the mechanisms of biomimetic mineralization of apatite [81]. These studies demonstrated the importance of the presence of ACP clusters in the pore wall of MBGs to accelerate surface reactions [85,88].

Preparation of 3D scaffolds based on the SiO_2_-P_2_O_5_ system with tailored meso-macroporosity followed [83]. Simultaneously, the group reported the improvement of MBGs as powders through the addition of inorganic ions such as Cerium, Gallium and Zinc to confer new biological properties [84].

Using state-of-the-art technology using scanning electron microscopy (SEM), solid-state NMR and X-ray diffraction (XRD), a method to quantify the formation of apatite and ion leaching from MBGs under in vivo conditions was proposed [89]. Additionally of note, new composites containing nanocrystalline apatite nuclei embedded in MBGs were designed at this time [87].

In the following years, María Vallet-Regí’s group progressed in the development and biological characterization of MBG scaffolds containing various therapeutic cations [90,95,96]. At the same time, the loading and release of various molecules of interest in orthopedic surgery, such as the antibiotics levofloxacin [91] and curcumin, were investigated using MBG as biomaterial [92]. Further structural studies performed by the group at this stage emphasize the importance of the spatial distribution of phosphate ions in the bioactive silicate-based glasses [93], and the need of controlling the hierarchical features of MBGs from the macroscopic to the nanoscopic scale with the perspective of their putative clinical application [124].

More recently, María Vallet-Regí has continued to use solid-state NMR to further explore the dependence of the composition of apatite formation on the surface properties of the glasses [98,100]. Moreover, the interaction of MBGs with cells, by investigating the in vitro colonization of pre-osteoblastic cells in stratified reactive scaffolds, was reported [99].

In 2017, the sequential release of several antimicrobial agents (rifampin, levofloxacin and vancomycin) against bacterial systems from hierarchical 3D multidrug scaffolds based on nanocomposite MBG bioceramic and polyvinyl alcohol (PVA) was investigated [101]. Levofloxacin was loaded into the mesopores of the nanocomposite, vancomycin was localized into the PVA biopolymer part and rifampin was loaded in the external coating. In addition, as a novel advance in the field of silica mesoporous nanoparticles, they proposed the use of such nanoparticles for the controlled release of four active components in a polypill [125]. In this regard, the group examined the effect of incorporating ions such as phosphorus [102] into MBGs and copper [103] into bioactive glass nanoparticles to improve their biological properties. Moreover, zwitterionic surfaces of MBG by functionalization with APTS and lysine were designed and tested for the prevention of bacterial adhesion [105]. Within the same time period, Vallet-Regí´s group also reported the use of molecular gates (e.g., pH-sensitive) in MBG scaffolds, analogous to those previously proposed for mesoporous silica materials, with the aim of treating bone infection and bone tumors [106,113]. In this work, innovative nanodevices based on the implementation of adenosine triphosphate (ATP) and ε-poly-l-lysine molecular gates using an MBG as support were developed. The systems worked properly with the antibiotic levofloxacin and the antitumoral drug doxorubicin.

At this stage of research development, the group also reported the development of several gallium-containing MBGs with improvements in their composition to favor the release of gallium ions to the medium to exert bactericidal action [107]. Further in vitro studies were also reportedly investigating the bioactivity of MBGs using osteoblast, osteoblast, and macrophages cultures [109,110]. In this respect, in vitro and in vivo studies using MBG scaffolds doped with Zn, an osteogenic and bactericidal element, in combination with the osteoinductor peptide osteostatin were recently carried out using different osteoblastic cell cultures and a rabbit femoral defect, respectively [68,72,73] Moreover, as part of the biological characterization of ion-doped BGs, the induction of VEGF secretion from mesoporous BG containing CuO and SrO and seeded with the bone marrow stromal cell line ST-2 was investigated [111]. The group has also proceeded to explore the effects of calcium and silicon mesoporous nanoparticles loaded with ipriflavone in osteoblast and osteoclast cultures [119], as well as the anti-bacterial efficiency of MBGs equipped with molecular gates for controlled antibiotic release [113]. In this work, MBG was functionalized with polyamines and capped with ATP as a molecular gate for the controlled release of the antibiotic levofloxacin. Phosphate bonds of the ATP are hydrolyzed in the presence of acid phosphatase, which significantly increases its concentration in bone infection at the same time that osteoinduction is favored.

Some recent studies of María Vallet-Regí´s research group include:
The use of combined MBG/polycaprolactone scaffolds to promote bone regeneration in an osteoporotic sheep model [114];The incorporation of Cerium in two oxidation states (III) and (IV) to beads of MBGs and alginates showing activity against reactive oxygen species [115];The effect of biomimetic biomineralization of MBG scaffolds on their physical properties and ability to favor osteogenic cell differentiation [116];The design of advanced Sr-doped MBG materials with zwitterionic surfaces [117] and MBGs containing copper [66] as putative bone implants;The investigation of the efficacy of MBGs containing 4% ZnO and four antibiotics (ciprofloxacin, levofloxacin, vancomycin and gentamicin) against E. coli and S. aureus bacterial strains [68];The evaluation of new processing methods of MBG scaffolds, namely using a negative PLA mold that is then extracted [118];The synthesis of mesoporous nanospheres loaded with ipriflavone for periodontal treatment [119], and their bioactivity tested on the differentiation of endothelial cells and macrophage maturation [121];The increase in the bone-forming ability of MBG-PCL composite scaffolds. In said scaffolds, microporosity was created by porogen removal, while 3D printing imparted macroporosity, and the MBG particles were responsible for the mesoporosity [120];The comparative evaluation of the effects of MBG particles and mesoporous nanospheres on RAW 264.7 and J774A.1 macrophage. Both materials allow the appropriated development and function of macrophages and do not induce polarization towards the M1 pro-inflammatory phenotype [122].

In summary, we can say that the field of MBGs for bone regeneration is a very current field that since their discovery 15 years ago has experienced a continuous growth that in recent years tends to accelerate. As was seen in this review article, there are many research groups interested in the subject, but the role played by Prof. María Vallet-Regí has been very prominent, as shown in Figure 5, which highlights some of her main achievements and how she is contributing to the booming and maturation of this subject.

## 6. Future Prospects

Looking at the upper part of the tree in Figure 5, we can highlight some of the current more advanced research lines that will arouse great interest in the coming years for applications in bone regeneration. For example, the development of stimulus-response systems that allow the design of molecular gates to control the release of substances contained within the pores of MBGs. Another aspect that is arousing great interest in recent years is the design of mesoporous nanoparticles [40,126]. Very important will also be the development of new animal models, both in healthy and osteoporotic animals for the in vivo evaluation of MBGs or the functionalization zwitterionization of the surfaces to make them more friendly to bone cells and increase their resistance to bacterial colonization, preventing and treating infection processes. Therapeutic ions with increasingly better understood and utilized biological action will also continue to be incorporated into MBGs.

Moreover, other research groups are investigating additional features of MBGs, for example, their angiogenic properties [20] or their possible use for the diagnosis and treatment of cancer [127]. As an example of the current importance of MBGs in the area of BGs, one can mention the book by Arcos and Vallet-Regí of 2020 [128]. The title is *“Bioactive glasses”* but four of the ten chapters are dedicated exclusively to MBGs. Furthermore, three other chapters coming also described aspects of MBGs in broader settings. Another proof of the current interest in MBGs is a large number of very recent reviews and research articles on specific aspects of these materials, such as the biological effect of doping with certain types of inorganic ions, in vitro assays and others [46,129,130,131,132].

Other foreseeable future developments: Several microRNA analogs (miRNA) and microRNA inhibitors (antagomir) were identified as regulators (stimulators or inhibitors) of osteoblastic growth and function [133]. In fact, several plasma miRNAs correlate with bone mineral density and could therefore be considered as an alternative to classical markers of bone remodeling in osteoporotic patients. Moreover, despite the rapid pharmacokinetics of these molecules in biological fluids, their stabilization in modified nanomaterials—such as MBGs—could be a novel therapeutic strategy to promote bone regeneration and repair [133].

## 7. Conclusions

This review is a tribute to the excellent work and outstanding achievements performed by the group led by Prof. María Vallet-Regí, making her a distinguished researcher in the field. Specifically, she introduced the concept of drug incorporation into the pores of mesoporous silica materials. Regarding the development of mesoporous glasses, the group led by Prof. María Vallet-Regí had a multidisciplinary and holistic approach including their synthesis and structural characterization by using state-of-the-art technology such as TEM or NMR spectroscopy and many others. This allowed the team to produce a complete knowledge of the materials investigated prior to addressing their in vitro and in vivo characterization. As can be seen in Figure 5, the tree of the MBGs after her contributions has bloomed lush and will continue to grow. We should emphasize that Prof. María Vallet-Regí must take credit for the school she created and that will be her legacy. After a complete introduction of the aspects related to mesoporous glasses enriched with bioactive agents for bone repair, the present review brings a synthesis of the discoveries made by Prof. María Vallet-Regí’s group, which constitute the most important findings during almost 20 years of research and innovations for multiple medical applications. 

## Figures and Tables

**Figure 1 pharmaceutics-14-00202-f001:**
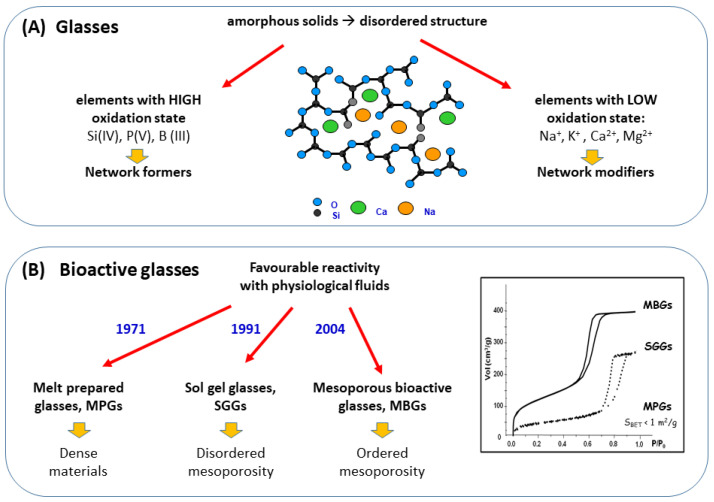
Main features of: (**A**) glasses and (**B**) the three families of bioactive glasses.

**Figure 2 pharmaceutics-14-00202-f002:**
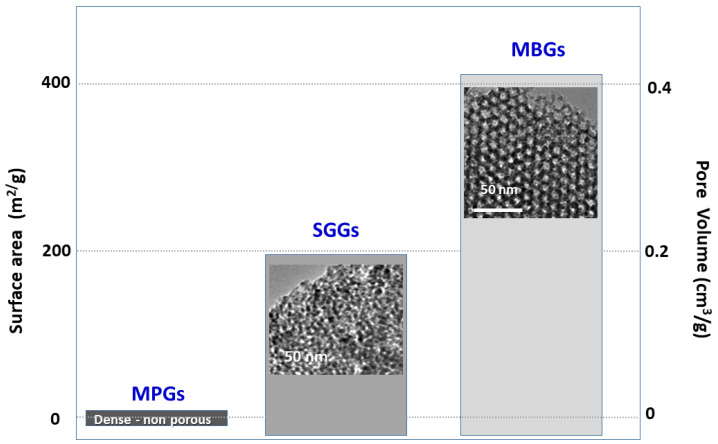
Textural properties of the three families of bioactive glasses. Transmission Electron Micrographs of the porous SGG and MBG are also included.

**Figure 3 pharmaceutics-14-00202-f003:**
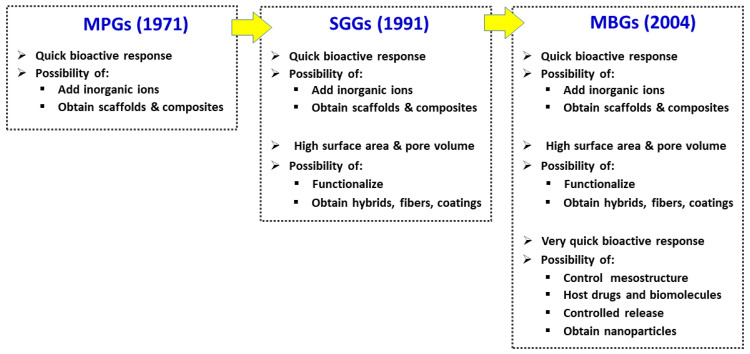
Main properties of the three families of BGs for bone regeneration applications.

**Figure 4 pharmaceutics-14-00202-f004:**
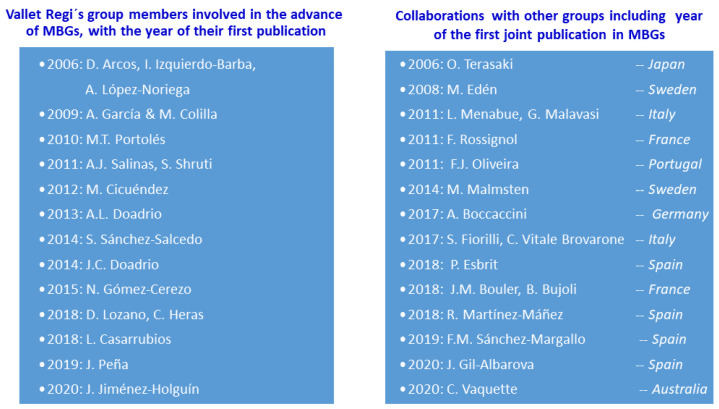
Members of the Vallet-Regí group and external collaborations in the field of MBGs.

**Figure 5 pharmaceutics-14-00202-f005:**
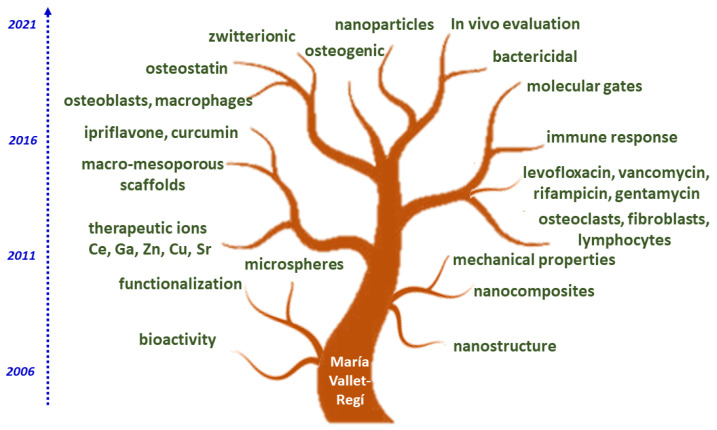
MBGs tree after María Vallet-Regí.

**Table 1 pharmaceutics-14-00202-t001:** Papers of Vallet-Regí from 2006 to 2021 including significant advances in MBGs.

Year	Features of MBGs Reported	Ref
2006	Huge textural properties, very quick bioactive response, different mesoporous arrangements	[35]
2008	MBG accelerated bioactivity mechanism; characterize 3D bicontinuous cubic network	[74]
	P location at the MBG structure and what happens when there is an excess of Ca	[75]
2009	Incorporation of P in mesostructured silicas to reduce the SiO_2_ leaching in water	[76]
	Essential role of calcium phosphate heterogeneities; solvent evaporation T controls mesoporous order	[77]
	Ordered mesoporous microspheres for bone grafting and drug delivery	[78]
	Mesoporous microspheres with doubly ordered core-shell structure	[79]
2010	Functionalizing MBGs for delivery of the anti-osteoporotic drug ipriflavone	[80]
	Biomimetic apatite mineralization mechanisms of MBGs as probed by^31^P, ^29^Si, ^23^Na and ^13^C NMR	[81]
	Interaction of MBGs with osteoblasts, fibroblasts and lymphocytes demonstrating biocompatibility	[82]
2011	Preparation of 3D scaffolds in the SiO_2_–P_2_O_5_ system with tailored meso-macroporosity	[83]
	Substitutions of cerium, gallium and zinc ions in SiO_2_–CaO–P_2_O_5_ MBGs	[84]
	^31^P and ^1^H NMR of amorphous and crystalline calcium phosphates grown biomimetically from MBGs	[85]
	Mechanical reinforcement of NMR scaffolds by a biomimetic process	[86]
2012	Nanocomposite with nanocrystalline apatite embedded into MBG	[87]
	Local structures of MBGs and their surface alterations in vitro: inferences from solid-state NMR	[88]
	Quantifying apatite formation and cation leaching from MBGs in vitro by using SEM, NMR XRD	[89]
2013	MBG scaffolds including cerium, gallium and zinc ions	[90]
	Biocompatibility and levofloxacin delivery of mesoporous materials	[91]
	Curcumin release from Cerium, Gallium and Zinc containing MBG scaffolds	[92]
	Probing of the spatial distribution of phosphate ions in MBGs by solid-state NMR	[93]
2014	Tailoring hierarchical meso-macroporous scaffolds from nanometric to macrometric scales	[94]
	In vitro antibacterial capacity and cytocompatibility of ZnO-enriched MBG scaffolds	[95]
	Effects of 3D nanocomposite bioceramic scaffolds on immune response	[96]
2015	Tailoring the biological response of mesoporous bioactive materials	[97]
	Composition-dependent in vitro apatite formation at MBG-surfaces quantified by NMR and XRD	[98]
2016	In vitro colonization of stratified bioactive scaffolds by preosteoblast cells	[99]
	Surface reactions of MBG monitored by solid-state NMR: concentration effects in SBF	[100]
2017	3D scaffold nanoapatite/MBG composite with multidrug sequential release against bacteria biofilm	[101]
	Structural characteristics of Sr-, Cu- and Co-doped MBGs influenced by the presence of P_2_O_5_	[102]
	Cu-containing MBG nanoparticles as multifunctional agents for bone regeneration	[103]
	Proton environments in biomimetic calcium phosphates formed in vitro from CaO–SiO_2_–P_2_O_5_ MBGs	[104]
	Prevention of bacterial adhesion to zwitterionic biocompatible MBGs	[105]
	Molecular gates in MBGs for the treatment of bone tumors and infection	[106]
2018	Highly-bioreactive silica-based MBGs enriched with gallium(III)	[107]
	Multifunctional scaffolds, pH-sensitive, for treatment and prevention of bone infection	[108]
	Effects of a MBG on osteoblasts, osteoclasts and macrophages	[109]
	Response of pre-osteoblasts and osteoclasts to Ga-containing MBGs	[110]
	Osteogenic effect of ZnO-MBGs loaded with osteostatin.	[71]
	VEGF secretion from bone marrow stromal cells by dissolution of glass particles containing CuO or SrO	[111]
	Effects of mesoporous SiO_2_–CaO nanospheres with ipriflavone on osteoblast/osteoclast co-cultures	[112]
	MBGs equipped with stimuli-responsive molecular gates controlled delivery of levofloxacin	[113]
2019	Osteostatin potentiates MBG scaffolds containing Zn^2+^ ions in human mesenchymal stem cells	[72]
	MBG/Ɛ-polycaprolactone scaffolds promote bone regeneration in osteoporotic sheep	[114]
	Ce(III) and (IV)-MBG/alginate beads: bioactivity, biocompatibility and reactive oxygen species activity	[115]
2020	Effect of biomimetic mineralization of MBG scaffolds on physical properties and in vitro osteogenicity	[116]
	Sr-releasing MBGs with anti-adhesive zwitterionic surface for bone regeneration	[117]
	Multifunctional antibiotic- and Zinc-containing MBG scaffolds to fight bone infection	[68]
	Development and evaluation of Cu-containing MBGs for bone defects therapy	[66]
	ZnO-MBG scaffolds loaded with osteostatin and mesenchymal cells in a rabbit bone defect in femur	[73]
	SrO-modified scaffolds based on MBGs/Polyvinyl alcohol composites for bone regeneration	[118]
	Ipriflavone-loaded mesoporous nanospheres with potential applications for periodontal treatment	[119]
2021	Multiscale porosity in 58S MBG/Polycaprolactone 3D-printed scaffolds for bone regeneration	[120]
	Effects of ipriflavone-mesoporous nanospheres on endothelial cells and modulation by macrophages	[121]
	Response of macrophages to particles and nanoparticles of an MBG: a comparative study	[122]

## Data Availability

Not applicable.

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
