# Peer review of "Mesoporous Bioglasses Enriched with Bioactive Agents for Bone Repair, with a Special Highlight of María Vallet-Regí’s Contribution"

_pharmaceutics, 2022, doi:10.3390/pharmaceutics14010202_

Round 1
Reviewer 1 Report
The argument for this review is a tribute to M Vallet-Regi who is a major actor in the firel of bone regeneration, due to its research on mesoporous bioactivated glasses.
A quick search on medline indicate that a review about M Vallet-Regi is already published in Pharmaceutics. It is my opinion the editor should check if this paper is not redondant.
The authors reviewed the history of mesoporous material as used for bone regeneration. They present the different steps the scientists followed from melt prepared glasses to mesoporous bioactived glasses. They compared the three main glasses for their structures and properties.
M Vallet-Regi published more than 300 papers, around 50 being about mesoporous bioactivated glasses, the authors gathered all this papers in one table. They participated to some of those. Papers about mesoporous bioactivated glasses are about 400. The authors quoted 133 papers. It would be interesting to know why they choose not to quote so many papers.
First sentence in the rationale is "Writing a review article in tribute to Maria Vallet-Regis’s scientific career is a colossal task.". I agree, and to help to be sure that this task is fulfilled correctly, I hardly suggest the authors to follow the PRISMA methods adapted for reviews, and to show it. (ref : Tricco, AC, Lillie, E, Zarin, W, O'Brien, KK, Colquhoun, H, Levac, D, Moher, D, Peters, MD, Horsley, T, Weeks, L, Hempel, S et al. PRISMA extension for scoping reviews (PRISMA-ScR): checklist and explanation. Ann Intern Med. 2018,169(7):467-473. doi:10.7326/M18-0850.). This will assess perfectly the quality of the review. It is a major revision, as it willadd a methods parts to the paper, but it will improve greatly its quality.
Author Response
Answers to Reviewer 1
The argument for this review is a tribute to M Vallet-Regi who is a major actor in the firel of bone regeneration, due to its research on mesoporous bioactivated glasses.
1) A quick search on medline indicate that a review about M Vallet-Regi is already published in Pharmaceutics. It is my opinion the editor should check if this paper is not redondant.
ANSWER: Thanks for the comment. The review mentioned refers to mesoporous silica compounds, i.e., materials based on pure SiO2, while our present review refers to mesoporous glasses, which in addition to SiO2 contain significant amounts of other oxides, particularly CaO and P2O5 and, therefore, they exhibit quite different properties and clinical applications than mesoporous silica compounds. Consequently, it is not redundant at all.
2) The authors reviewed the history of mesoporous material as used for bone regeneration. They present the different steps the scientists followed from melt prepared glasses to mesoporous bioactived glasses. They compared the three main glasses for their structures and properties.
M Vallet-Regi published more than 300 papers, around 50 being about mesoporous bioactivated glasses, the authors gathered all this papers in one table. They participated to some of those. Papers about mesoporous bioactivated glasses are about 400. The authors quoted 133 papers. It would be interesting to know why they choose not to quote so many papers.
ANSWER: The present review forms a part of the special issue of Pharmaceutics launched to commemorate 20 years since Maria Vallet-Regi proposed in 2001 pure SiO2 mesoporous materials for medical applications. In 2004, emerged a branch of these materials, namely mesoporous bioactive glasses which, in addition to SiO2, contain oxides of other elements, mainly CaO and/or P2O5 and that exhibit somewhat lower surface area and porosity but are more appropriate for bone regeneration applications because their bioactive behaviour. Of a total amount of almost 400 publications on mesoporous materials by Vallet-Regí et al., our present review has focused on those dealing with bioactive mesoporous glasses (close to 60 research papers). To put the subject in context, we have also included other outstanding references on the same topic from other research groups along with a dozen review articles and book chapters co-authored by Prof. María Vallet-Regí, in which mesoporous bioactive glasses were mentioned. The sum of these three groups of references gives the 133 references mentioned by the reviewer.
3) First sentence in the rationale is "Writing a review article in tribute to Maria Vallet-Regis’s scientific career is a colossal task.". I agree, and to help to be sure that this task is fulfilled correctly, I hardly suggest the authors to follow the PRISMA methods adapted for reviews, and to show it. (ref : Tricco, AC, Lillie, E, Zarin, W, O'Brien, KK, Colquhoun, H, Levac, D, Moher, D, Peters, MD, Horsley, T, Weeks, L, Hempel, S et al. PRISMA extension for scoping reviews (PRISMA-ScR): checklist and explanation. Ann Intern Med. 2018,169(7):467-473. doi:10.7326/M18-0850.). This will assess perfectly the quality of the review. It is a major revision, as it will add a methods parts to the paper, but it will improve greatly its quality.
ANSWER: We appreciate the reviewer's suggestion regarding the PRISMA methods adapted for the reviews. However, the present article is a very special review aiming at emphasizing the scientific contribution of María Vallet Regí's studies on bioactive mesoporous glasses since her first publication on the subject in 2006. In the same period, she has published many other articles concerning other types of mesoporous materials that have not been considered in this review. During the last 15 years, other authors have also published articles on this subject, but we have only included those considered more relevant by the authors’ opinion. To clarify that, we decided to include at the section 1 of the article “Rationale and Objectives” the following sentences:
"This review focuses on all the articles published by María Vallet-Regí et al. on bioactive mesoporous glasses, namely those containing CaO or P2O5 together with SiO2 in their composition, obtained by using surfactant acting as structure directing agents. … To add context to these contributions, outstanding references by other authors on the same topic were also included. "
Reviewer 2 Report
The analyzed review is a tribute to a great researcher in the field-Prof. María Vallet-Regi. Making a wide introduction in the aspects related to mesoporous bioglasses enriched with bioactive agents for bone repair, the manuscript brings a synthesis of the discoveries made by Prof. María Vallet-Regi, the most important researches during the almost 20 years of researches and innovations with multiple medical applications. The review article is well done, has a logical development, and the contribution of Prof. María Vallet-Regi is correlated with the school on that she created it and left behind, which is very important in the life and career of a researcher of such magnitude.
As an observation on my part - a revision of the English language and an increased attention to the bibliographic indications introduced, respectively to be in agreement with the requirements of the Guide and of the instructions for the authors.
Author Response
Answers to Reviewer 2
The analyzed review is a tribute to a great researcher in the field-Prof. María Vallet-Regi. Making a wide introduction in the aspects related to mesoporous bioglasses enriched with bioactive agents for bone repair, the manuscript brings a synthesis of the discoveries made by Prof. María Vallet-Regi, the most important researches during the almost 20 years of researches and innovations with multiple medical applications. The review article is well done, has a logical development, and the contribution of Prof. María Vallet-Regi is correlated with the school on that she created it and left behind, which is very important in the life and career of a researcher of such magnitude.
1) As an observation on my part - a revision of the English language and an increased attention to the bibliographic indications introduced, respectively to be in agreement with the requirements of the Guide and of the instructions for the authors.
ANSWER: We are thankful to the reviewer for their comments on the quality of our review. The English language and the bibliography have been revised.
Reviewer 3 Report
In this paper an effort is made to describe the huge work of Vallet-Regí in the area of MBGs. Below find some comments and suggestions towards the overall improvement of the manuscript:
- “In the 1990s, while SGGs bioceramics were been developed, in the field of catalysis, mesoporous silica materials were designed to increase the pore size of zeolites”:
-were been: please remove been
-SGGs bioceramics: they are glasses and not ceramics, so bioceramics should be removed
- “However, SGGs failed to fulfill the initial 97 expectations for clinical applications because their bioactive response was analogous 98 to MPGs”: what were the initial expectations that were not fulfilled with the SGGs? Please give some examples
- “However, melt-prepared BGs (MPGs), found only limited clinical applications, namely in the dental field, for replacing the middle ear chain ossicles, or as particulates in bone grafts, in the last twenty years. This gave rise, in the 1990s, to a new family of BGs, the so-called bioactive sol-gel glasses (SGGs)”: what gave rise? The limited clinical applications or other more attractive properties of SGGs like their higher textural properties that made them promising candidates for local drug delivery? I think the authors need to rephrase.
- Figure 2: I suggest the authors declare from what kind of SGGs and MBGs the TEM images are.
5.“This process produces advantageous textural properties as well as much faster in vitro bioactive response than that of the other BGs family members”: please add reference
- “However, this kind of MBG form still needs a great deal of research before it could be materialized into a clinical application”: please explain the reasons, what are their current disadvantages that make them need more research
- figure 4: my suggestion is to add near the names of the collaborators their current institutes and near the names of the group members their level of academic qualification, do not report only the PhD students
- Please check some minor syntax errors
- My most important comment is that in the conclusions section there should be a summary of the key features of the work of Vallet-Regí and her group that makes her a distinct researcher among the others in the field. In my mind for example she is one of the first to suggest incorporation of drugs into the pores of mesoporous silica materials and this is something that she has to take the credit for among others, and should be highlighted in the conclusions.
Author Response
Answers to Reviewer 3
In this paper an effort is made to describe the huge work of Vallet-Regí in the area of MBGs. Below find some comments and suggestions towards the overall improvement of the manuscript:
- “In the 1990s, while SGGs bioceramics were been developed, in the field of catalysis, mesoporous silica materials were designed to increase the pore size of zeolites”:
-were been: please remove been
-SGGs bioceramics: they are glasses and not ceramics, so bioceramics should be removed
ANSWER: The words “been” and “bioceramics” were removed from the corresponding sentence
- “However, SGGs failed to fulfill the initial 97 expectations for clinical applications because their bioactive response was analogous 98 to MPGs”: what were the initial expectations that were not fulfilled with the SGGs? Please give some examples
ANSWER: Because their high porosity and surface silanol groups, SGGs exhibited a bioactive response, i.e. were coated by an apatite-like layer after soaking in a Simulated Body Fluid (SBF), even for silica content of 90%; while such response was limited to a maximum content of 60 % for MPGs. These findings led to the assumption that SGGs might present a quicker bioactive response in SBF than MPGs. However, this was not the case; a similar period of time (3 to 7 days) was necessary to form the apatite-like layer with both types of glasses (SGGs and MPGs) in these assays. On the other hand, a considerable decrease of the time required to form the apatite-like layer was observed for MBGs, indicative of its superior bioactive response of this family of glasses.
This paragraph was rewritten:
“Indeed the bioactivity window was expanded in SGGs because they were coated by an apatite-like layer after soaking in a Simulated Body Fluid (SBF), even for silica contents of 90%; while such response was limited to a maximum content of 60 % for MPGs. These findings led to the assumption that SGGs might present a quicker bioactive response in SBF than MPGs. However, this was not the case; a similar period of time (2 to 7 days) was necessary to form the apatite-like layer in both types of glasses [25].”
- “However, melt-prepared BGs (MPGs), found only limited clinical applications, namely in the dental field, for replacing the middle ear chain ossicles, or as particulates in bone grafts, in the last twenty years. This gave rise, in the 1990s, to a new family of BGs, the so-called bioactive sol-gel glasses (SGGs)”: what gave rise? The limited clinical applications or other more attractive properties of SGGs like their higher textural properties that made them promising candidates for local drug delivery? I think the authors need to rephrase.
ANSWER: This paragraph has been deeply rewritten, eliminating some inaccuracies and better explaining some concepts that were not clear in the previous version of the manuscript.
“However, melt-prepared BGs (MPGs) found only rather few clinical applications, namely in the dental field, for replacing the middle ear chain ossicles, or as particulates in bone grafts, in the last twenty years. For this reason, in the 1990s, to a new family of BGs, the so-called bioactive sol-gel glasses (SGGs) was obtained by wet chemistry methods, exhibiting high porosity and specific surface area [31]. These interesting textural properties led to the assumption that SGGs could be used as matrices in drug delivery systems. Furthermore, the sol-gel method allowed processing BGs as coatings or fibers, which are shapes difficult to obtain with the traditional quenching of a melt meth-od. However, SGGs did not meet all the initial expectations for clinical applications because their bioactive response was analogous to MPGs and because due to their large pore size dispersion they did not allow efficient control of the release of biologically active substances.”
- Figure 2: I suggest the authors declare from what kind of SGGs and MBGs the TEM images are.
ANSWER: Figure 2 is rather qualitative since what it is trying to show is that MBGs have textural properties (surface area and porosity) double that of SGGs and that being both mesoporous materials in MBGs all pores are ordered and have the same size. Therefore, no details of both glasses were given in the previous version of the manuscript.
Following the reviewer's comment, the following sentence has been added:
The TEM images of SGG and MBG shown are glasses of the SiO2-CaO-P2O5 system. The difference lies in the synthesis method used: classical sol-gel method for the former and the latter in the presence of surfactants and self-assembly induced by solvent evaporation (EISA method). As shown in the Figure, the mesopores of the SGGs are disordered and present different sizes. In contrast, an ordered arrangement of pores of the same size is observed in MBGs.
5.“This process produces advantageous textural properties as well as much faster in vitro bioactive response than that of the other BGs family members”: please add reference
ANSWER: A reference has now been included [11].
- “However, this kind of MBG form still needs a great deal of research before it could be materialized into a clinical application”: please explain the reasons, what are their current disadvantages that make them need more research
ANSWER: The following explanation has been added to the revised version of the manuscript:
“Indeed, there is still scarce preclinical research in animal models with MBGs. The current results are very promising, but several aspects still need to be elucidated before considering their putative clinical use: the ideal composition of the glass, the optimal resorption rate, the type and amount of therapeutic ions included, or the addition of different bioactive drugs presenting osteogenic, bactericidal, angiogenic, anticancer, anti-inflammatory, antioxidant and/or antiviral properties.”
- figure 4: my suggestion is to add near the names of the collaborators their current institutes and near the names of the group members their level of academic qualification, do not report only the PhD students
ANSWER: Figure 4 was modified according this important comment of the reviewer.
On the left of the previous version of the figure, the researchers who were PhD students when they did research on bioactive mesoporous glasses were highlighted. There are 7 students of which 5 have already read their thesis and 2 are about to do so. To make clearer the Figure, in the revised manuscript we have removed the label (PhD student) from the figure and added an explanation in the text indicating the names of the PhD students who have completed their PhD theses on bioactive mesoporous glasses within María Vallet-Regí's group.
“We have also included the names of PhD students whose Theses have focused exclusively on MBGs since, despite being trainees, their participation is considered very important in the described developments. They are: A. López-Noriega, S. Shruti, M. Cicuéndez, N. Gómez-Cerezo, C. Heras, L. Casarrubios, J. Jiménez-Holguín.
As for the collaborations (right side of the figure), the modification proposed by the reviewer is complicated. Fifteen years have passed and in that time some of the Professors have retired and others have changed their affiliation. If what the reviewer proposes could be done rigorously, it would be ideal, but in several cases it would be necessary to decide which affiliation should be used: the one at the time of the collaboration or the current one, which in some cases has changed and in other cases they are retired. Indeed, all contributors are sufficiently recognized so that anyone can easily find them in any database. However, the nationality of the contributors can be stated and this is what has been included in the revised version of the figure.
- Please check some minor syntax errors
ANSWER: Syntaxis has been thoroughly revised and corrected when appropriate.
- My most important comment is that in the conclusions section there should be a summary of the key features of the work of Vallet-Regí and her group that makes her a distinct researcher among the others in the field. In my mind for example she is one of the first to suggest incorporation of drugs into the pores of mesoporous silica materials and this is something that she has to take the credit for among others, and should be highlighted in the conclusions.
ANSWER: We greatly appreciate this particular comment of the reviewer. Thus, the conclusion section has been rewritten in a separate section as follows:
“This review is a tribute to the excellent work and outstanding achievements performed by the group led by Prof. María Vallet-Regi, making her a distinguished researcher in the field. Specifically, she introduced the concept of drug incorporation into the pores of mesoporous silica materials. Regarding the development of mesoporous glasses, the group led by Prof. María Vallet-Regi had a multidisciplinary and holistic approach including their synthesis and structural characterization by using state-of-the-art technology such as TEM or NMR spectroscopy and many others. This allowed the team to produce a complete knowledge of the materials investigated prior to addressing their in vitro and in vivo characterization. As can be seen in Figure 5 the tree of the MBGs after her contributions has bloom lush and will continue to grow. We should emphasize that Prof. María Vallet-Regi must take credit for the school she created and that will be her legacy. After a complete introduction of the aspects related to mesoporous glasses enriched with bioactive agents for bone repair, the present review brings a synthesis of the discoveries made by Prof. María Vallet-Regi’s group, which constitute the most important findings during almost 20 years of research and innovations for multiple medical applications.”
Round 2
Reviewer 1 Report
Thanks for your answer
even if a prisma would have been better